# The unusual quadruple bonding of nitrogen in ThN

Zejie Fei[1,7], Jia-Qi Wang[2,3,7], Rulin Tang[4], Yuzhu Lu[4], Changcai Han[1], Yongtian Wang[1], Jing Hong[1,5], Changwu Dong[1], Han-Shi Hu [2], Xiao-Gen Xiong [6] ✉, Chuangang Ning[4] ✉, Hongtao Liu[1] ✉ & Jun Li [2]

Nitrogen has five valence electrons and can form a maximum of three shared electron-pair bonds to complete its octet, which suggests that its maximum bond order is three. With a joint anion photoelectron spectroscopy and quantum chemistry investigation, we report herein that nitrogen presents a quadruple bonding interaction with thorium in ThN. The quadruple Th≡N bond consists of two electron-sharing Th-N $\pi$ bonds formed between the Th-$6d_{xz}/6d_{yz}$ and N $2p_x/2p_y$ orbitals, one dative Th←N $\sigma$ bond and one weak Th←N $\sigma$ bonding interaction formed between Th-$6d_z^2$ and N $2s/2p_z$ orbitals. The ThC molecule has also been investigated and proven to have a similar bonding pattern as ThN. Nonetheless, due to one singly occupied σ-bond, ThC is assigned a bond order of 3.5. Moreover, ThC has a longer bond length as well as a lower vibrational frequency in comparison with ThN.

Chemical bond is defined as the interaction between two or a group of atoms to form molecules, clusters and solids[1]. Although chemical bond is one of the most fundamental concepts in chemistry, it is not a physically observable quantity, and cannot be directly measured. The single-, double- and triple-bond are nomenclatures of chemistry refer to chemical bonds between two atoms consisting of one, two and three pairs of electrons, respectively, holding these two atoms together[1]. Generally, a high bond order indicates that the two atoms are strongly bonded with short length and large stiffness, thus the existence of multiple bonding between two atoms is mainly investigated experimentally by exploring the bond length, bond dissociation energy and vibrational frequencies (or force constants), etc. It is known from the textbook that the maximum bond order between two main-group atoms is three. This is based on the postulation that the tetrahedral arrangement of the four pairs of electrons in the valence shells of the main-group element, so that a maximum of three pairs of electrons can form a bond when two tetrahedra are sharing a face. Recently, with the development of the high-level theoretical calculations, the possibility

of quadruple bonding between two carbon atoms has been discussed in $C_2$ molecule[2-5]. Nevertheless, the debate on this statement has been raised for several years due to an obvious drawback of quadruple bond in $C_2$ molecule, the unreasonable weak bond strength. For instance, the experimentally determined vibrational frequencies in $C_2$ molecule are even weaker than that of the stereotypical triple bond in $HC≡CH$[6]. Theoretical analysis also showed that fluorine can form quadruple bond with alkali earth and boron group elements through the significant charge donation from F into its neighboring atom's vacant valence orbitals[7]. Nitrogen has five valence electrons and is able to form four or five bonds in total with other atoms, such as in $NH_4^+$, $N_2O_5$ and $NO_3^-$, etc. However, the common knowledge of the maximum bond order between a nitrogen atom and a single other atom is still three. The typical nitrogen-participated triple-bonded molecules have been summarized in Pyykkö's review article[8]. To the best of our knowledge, quadruple bond involving nitrogen accompanied with combined experimental and theoretical evidences has not been discussed until now[9-11].

[1]Key Laboratory of Interfacial Physics and Technology, Shanghai Institute of Applied Physics, Chinese Academy of Sciences, Shanghai 201800, China. [2]Department of Chemistry, Tsinghua University, Beijing 100084, China. [3]College of Science, Beijing Forestry University, Beijing 100083, China. [4]Department of Physics, State Key Laboratory of Low Dimensional Quantum Physics, Collaborative Innovation Center of Quantum Matter, Tsinghua University, Beijing 100084, China. [5]University of Chinese Academy of Sciences, Beijing 100049, China. [6]Sino-French Institute of Nuclear Engineering and Technology, Sun Yat-sen University, Zhuhai 519082, China. [7]These authors contributed equally: Zejie Fei, Jia-Qi Wang. ✉e-mail: xiongxg@mail.sysu.edu.cn; ningcg@tsinghua.edu.cn; liuhongtao@sinap.ac.cn

Yet it is worth noting that carbon has been proven to possess the potential to form a C≡U quadruple bond, consisting of two σ-bonds and two π-bonds, in the CUO molecule[12]. Since the formation of one σ-bond and two π-bonds is the classic picture of triple bond, the second σ-bond has been denoted as one of the key issues in the unconventional quadruple bond between p-block element and heavy metal element, which has been rationalized by various approaches of modern chemical bonding analysis. Using advanced gas phase spectroscopy, we can experimentally assess the bond strength by measuring the vibrational frequencies of diatomic molecules. For example, with the help of anion photoelectron spectroscopy, the Rh≡B quadruple bond has been found and rationalized in RhB(BO)⁻ and RhB[13]. Later on, another boron-metal quadruple bond was found in BFe(CO)₃⁻ by infrared photodissociation spectroscopy[14]. Inspired by these previous reports, herein we explore the potential and bonding characteristics of nonmetallic elements, C and N, in the second period, to form a quadruple bond with heavy element Th in ThC and ThN molecules, respectively. Using the sophisticated quantum chemical method up to the level of CCSD(T) (coupled-cluster singles-and-doubles plus perturbative triples), the calculated bond length in ThC is comparable with $R_{(Th≡C)}$ predicted by the average triple bond radii, while the bond length in ThN is noticeably shorter than predicted $R_{(Th≡N)}$. In order to further study these two molecules, we carried out the cryogenically slow velocity map imaging (cryo-SEVI) experiments which, the anions are cryogenically cooled to temperatures as low as 5 K in a radiofrequency ion trap and the velocity of the photoelectron is a few cm⁻¹ above the photodetachment threshold[15], together with theoretical calculations, indicate high bond orders of 3.5 and 4 for ThC and ThN, respectively. Insights into the bonding characterize the formation of two electron-sharing Th-N/C π-bonds, one dative Th←N/C σ bond and one weak polarized Th←N/C σ bond corresponding to $2p_x$/$2p_y$ and $2s$/$2p_z$ orbitals of C/N interacting with the $6d_{xz}$/$6d_{yz}$ and $6d_{z^2}$ orbitals of Th, respectively.

## Results

### Photoelectron spectra of ThC⁻ and ThN⁻

The cryo-SEVI apparatus[15,16] applied in our experiment has been widely used in the study of photoelectron spectroscopy of atomic and molecular clusters[17–21]. The ThN⁻ and ThC⁻ anions were generated by laser ablation of a thorium metal disk in the presence of NF₃ and CH₄ gas, respectively. The anions were captured by an octupole radiofrequency (RF) ion trap and cooled through collisions with the buffer gas (20% H₂ and 80% He). The ion trap was mounted on the second stage of a liquid helium refrigerator with a tunable temperature in the range of 5K-300K[22,23]. Then, the anions were ejected out by the pulsed potentials on the end caps of the trap and analyzed using a Wiley-McLaren type time-of-flight (TOF) mass spectrometer[24,25]. The anion species of interest was mass-selected before being photo-detached by

a tunable laser. The photoelectron kinetic energies were measured with a SEVI system.

Figure 1 shows the photoelectron image and spectrum of ThC⁻ at 649 nm (photon energy 1.909 eV) using the cryo-SEVI apparatus. Peak X represents the transition from the ground state of ThC⁻ to that of ThC, yielding an accurate first adiabatic detachment energy (ADE₁) to be 1.562 eV for ThC⁻, which also represents the electron affinity (EA) of the neutral ThC. The peak X is followed by three peaks a, b and d, corresponding to the ν = 1, 2 and 3 vibrations with binding energies of 1.663 eV, 1.763 eV and 1.863 eV, respectively, with an almost equal spacing around 0.1 eV. These four peaks should correspond to the ground-state vibrational progression with a frequency of 815 cm⁻¹. Peak A at 1.693 eV corresponds to the transition from the ground state of ThC⁻ to the first electronically excited state of ThC, and this binding energy is defined as the second adiabatic detachment energy (ADE₂) for ThC⁻. Peak c (ν′ = 1) at 1.779 eV is followed by peak d (ν′ = 2). These three peaks (A, c and d) were separated equally by 0.088 eV, which should correspond to the first excited-state vibrational progression with a frequency of 710 cm⁻¹. The relative intensities of these peaks also suggest that they represent a vibrational progression of a electronic state. As should be noted, the intensity of peak d was stronger than the normal, since the two above-mentioned vibrational progressions were overlapped in this binding energy region. Besides, a very weak vibrational hotband peak (label as hb in Fig. 1) was observed at 1.474 eV, due to the imperfect anion cooling. The anion vibrational frenquecy of 710 cm⁻¹ for ThC⁻ anion can be estimated from the energy difference between the peaks hb and X.

The photoelectron spectrum of ThN⁻ displayed in Fig. 2 was also taken on the cryo-SEVI apparatus. The lowest binding energy band X corresponds to the detachment transition from the ground state of ThN⁻ to that of neutral ThN, generates whereas the near equally spacing binding energy bands X, a and b, indicating the vibrational progression with ν = 0, 1 and 2, respectively, for the ground-state transition. the fundamental vibrational frequency of 944 cm⁻¹ was obtained by fitting the two vibrational interval values with anharmonicities. Band X yielded the ADE of ThN⁻ to be 1.576 eV, which also represents the EA of the neutral ThN. The binding energies of peaks a and b are 1.693 eV and 1.809 eV, respectively. The photoelectron spectra of ThC⁻ and ThN⁻ are served as the electronic-state fingerprints to allow analyses of their structures and bonding by comparison with the theoretical calculations. The observed PES features and their binding energies were compared with the theoretical results as shown in Tables 1 and 2 for ThC⁻ and ThN⁻, respectively.

Abundant accurate vibrational features were obtained through the experimental spectra. For ThC, ν = 0, 1, 2 and 3 levels and ν′ = 0, 1 and 2 of the ground state and first excited state were observed, respectively. The wavenumbers $ΔG_{ν+1/2}$ for the transitions between vibrational levels labeled as ν + 1 and ν can be calculated using

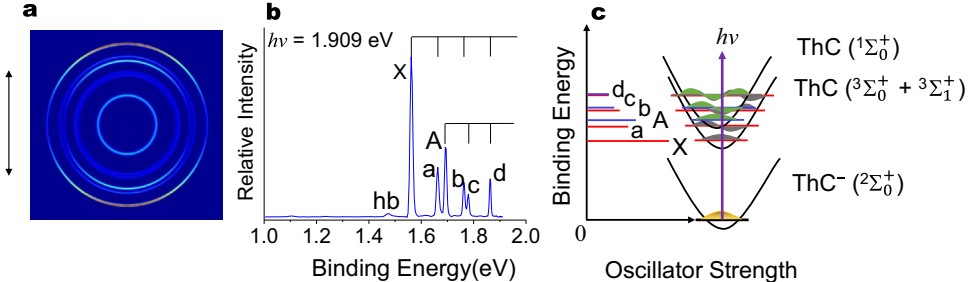

**Fig. 1 | Photoelectron spectrum of ThC⁻. a** Photoelectron images, **b** spectra for ThC⁻ at photon energy hν = 1.909 eV and **c** the energy level Schematic for the ground state ThC⁻ ($^2Σ_0^+$) → ThC ($^3Σ_0^+ + ^3Σ_1^+$) transition and first excited state ThC⁻ ($^2Σ_0^+$) → ThC ($^1Σ_0^+$) transition processes, the transitions are labelled to match the corresponding peaks in **b**. The vertical lines in **b** represent vibrational structures, and the double arrow on the left of the image in **a** indicates the laser polarization. Source data are provided as a Source data file.

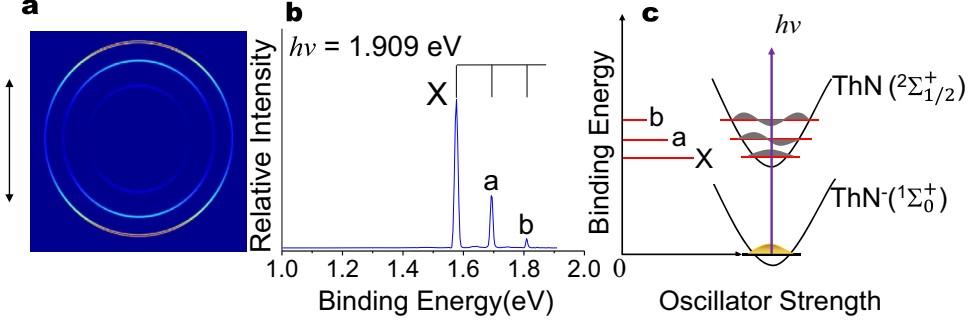

**Fig. 2 | Photoelectron spectrum of ThN⁻. a** Photoelectron images, **b** spectra for ThN⁻ at photon energy $hv = 1.909$ eV and **c** the energy level schematic for the ground state ThN⁻ ($^1\Sigma_0^+$) → ThN ($^2\Sigma_{1/2}^+$) transition process, the transitions are labelled to match the corresponding peaks in **b**. Vertical lines in **b** represent vibrational structures, and the double arrow on the left of the image in **a** indicates the laser polarization. Source data are provided as a Source data file.

formula (1)[26], where $\omega_e$ is the vibrational wavenumber of the classical harmonic oscillator and $\omega_e x_e$ is the anharmonic constant. In our study, the molecular constants of $\Delta G_{1/2} = 815$ cm⁻¹, $\omega_e = 824$ cm⁻¹ and $\omega_e x_e = 4.5$ cm⁻¹ were obtained for the ground state of ThC, and the $\Delta G_{1/2}' = 694$ cm⁻¹, $\omega_e' = 710$ cm⁻¹ and $\omega_e' x_e' = 8$ cm⁻¹ were obtained for first excited state of ThC. For ThN, one additional $v = 2$ vibrational feature was observed other than the previous study ($v = 0$ and 1)[9]. The first vibrational interval $\Delta G_{1/2} = 944$ cm⁻¹ of the first two peaks is in good agreement with the 950(15) cm⁻¹, reported by Heaven's group using dispersed fluorescence spectrum[8,9]. Furthermore, some other molecular constants of $\omega_e = 952$ cm⁻¹ and $\omega_e x_e = 4$ cm⁻¹ were also obtained for the ground state of ThN.

$$\Delta G_{v+1/2} = G(v+1) - G(v) = \omega_e - \omega_e x_e - 2\omega_e x_e v \tag{1}$$

**Theoretical bond lengths, vibrational frequencies and detachment energies**

Theoretical calculations were performed using relativistic quantum chemical methods to obtain the ground and low-lying excited states of the studied molecules, and the methods used in our calculations are explicitly stated in the Supplementary Information (SI). The diatomic bond lengths were fully optimized at the CCSD(T)[27,28] level and the results are listed in Supplementary Table 1 of SI. The optimized geometries at the level of CCSD(T) accompanied with quadruple-ζ basis sets (AVQZ for C and N, VQZ-PP for thorium) reveal that ThC⁻ has a $^2\Sigma$ ground state with $R_{(Th-C)} = 1.985$ Å. The detachment of one electron from the anion will produce the $^3\Sigma$ neutral ground state with a shortened Th-C bond length of 1.948 Å. For the diatomic thorium nitride, the $R_{(Th-N)}$ at the CCSD(T) level were 1.854 and 1.820 Å for anion and neutral molecules, with the corresponding ground states of $^1\Sigma$ and $^2\Sigma$, respectively. The vibrational frequency calculations were also carried out at the CCSD(T) level and the results were 847 cm⁻¹ and 960 cm⁻¹ for ThC and ThN, respectively.

The theoretical ADE was calculated as the energy difference between the ground states of the anion and neutral in which the zero-point energy (ZPE) corrections were included. Our CCSD(T) calculations suggest that the ADE results are 1.52 and 1.55 eV for ThC⁻ and ThN⁻, respectively. Due to the strong relativistic effects of actinide elements, the reliable theoretical prediction of the electronic structures of molecules containing Th requires a thorough incorporation of both scalar and spin-orbit (SO) effects. The low-lying excited states using a spin-orbit complete-active-space second-order perturbation (SO-CASPT2) approach[29–31] are detailed presented in Supplementary Tables 2 and 3 for ThC and ThN, respectively. The final state corresponding to the peak B of the experimental spectrum shown in Fig. 1 is tentatively assigned to the second excited state of ThC with the calculated ADE₂ value of 1.66 eV as listed in Table 1 and Supplementary Table 2, while the first excited state has a very close energy with the

ground state and we assign the peak X to be superposition of the transition from ground state of ThC⁻ to the ground state and first excited state of ThC. Based on the results shown in Tables 1 and 2, our calculated ADEs are consistent with the experimental data, which validates the robust of using SO-CASPT2 approach to quantitatively predict the correct energies of low-lying excited states for molecules containing Th.

## Discussion

### Bonding orbital analysis

Except for rare cases[32], bond length generally has a negative correlation with the bonding strength between two certain atoms. The average additive covalent bond radii have been comprehensively compiled for almost all the elements in the entire periodic table by Pyykkö, etc[8,33–35]. As a comparison, our optimized $R_{(Th-C)}$ in ThC is slightly shorter than the Pyykkö's average triple bond length $R_{(Th\equiv C)}$ (~ 1.96 Å), while the $R_{(Th-N)}$ in ThN is noticeably shorter than the average $R_{(Th\equiv N)}$, which is around 1.90 Å. The anomalous short bond length and large vibrational frequency of ThN impel us to thoroughly investigate the nature of the chemical bond between the two atoms in the studied thorium-containing diatomic molecules.

The Kohn-Sham MOs and the orbital diagrams of ThC and ThN are shown in Supplementary Figure 1 and Fig. 3, respectively. As seen from the two figures and evidenced by the symmetrized fragment orbital (SFO) populations in Supplementary Table 4, the molecular orbitals $10\pi$ and $18\sigma$ can be verified as the bonding orbitals between Th and N/C, which contribute bond order 3 and 2.5 for ThN and ThC, respectively. Thorium prefers binding carbon and nitrogen via the formation of degenerate $\pi$-bonding orbitals, in which two un-paired Th ($6d_{xz}$, $6d_{yz}$) electrons are involved. $18\sigma$ of ThC is a bonding orbital stemming from the combination of the C $2p$ orbital and $7p6d5f$-hybridized AOs of Th. The bonding pattern of ThC derived from our DFT calculations shows the neutral molecule has a high diradical character, indicating the diatomic molecule a reactive species. During the formation of thorium carbide and nitride, only one of the two Th $7s$ electrons is involved in the bonding formation, resulting in the remaining one $7s$ electron occupying the non-bonding singly-occupied MO $19\sigma$, which has a conspicuous $7s$ character. Apart from the bonding orbitals $10\pi$ and $18\sigma$, with respect to the $2s$ atomic orbital of C/N, the MO $17\sigma$ has been significantly destabilized when compared with the atomic orbital energy during the bond formation. SFO population analyses displayed in Supplementary Table 5 show that $17\sigma$ has around 10% Th-$6d_z^2$ contribution. Besides, thorium has four valence electrons, and the strong direct and indirect relativistic effects leading to the small gap between the $6d$ and $7s$ atomic shells. The CCSD(T)/VTZ-PP calculations show that exciting an electron from $7s$ to $6d$ AO, resulting in an excited state with 4 unpaired electrons with configuration as [Rn] $(7s)^1(6d)^3$, costs only 13.2 kcal/mol, which can be easily compensated by the thermal of the exothermic reaction. Regarding the delocalized

**Table 1 | The experimental and theoretical adiabatic detachment energies (ADEs) of ThC⁻, the vibrational frequencies (in cm⁻¹) and the corresponding electronic states of ThC⁻ and ThC**

| Peak | Peak position (exp.) | ADEs (exp.) | ADEs (theo.) | Freq[a] (exp.) | Freq (theo.) | Final SO state |
|------|----------------------|-------------|--------------|----------------|--------------|----------------|
| hb | 1.474 (1) | | | 710 | 773 | |
| X | 1.562 (1) | 1.562(1) | 1.52 (ADE₁) | 815 | 847 | $^3\Sigma_0^+$ + $^3\Sigma_1^+$ |
| a | 1.663 (1) | | | | | |
| A | 1.693 (1) | 1.692(1) | 1.66 (ADE₂) | 694 | | $^1\Sigma_0^+$ |
| b | 1.763 (1) | | | | | |
| c | 1.779 (1) | | | | | |
| d | 1.863 (1) | | | | | |

The theoretical ADE₁ was calculated at the CCSD(T) level. The energy difference between second vertical detachment energy (VDE₂) and first vertical detachment energy (VDE₁) calculated at the SO-CASPT2 level was added to ADE₁ to generate the ADE₂. The vibrational frequency calculations were performed at the CCSD(T) level. AVQZ and VQZ-PP basis sets were used for C and Th, respectively, for all CCSD(T) calculations, while AVTZ and VTZ-PP basis sets were used in SO-CASPT2 calculation. All energies are in eV and all vibrational frequencies are in cm⁻¹.
[a]The value of the first vibration interval $\Delta G_{1/2}$.

**Table 2 | The experimental and computational ADE (in eV) of ThN⁻, the vibrational frequencies (in cm⁻¹) and the corresponding electronic states of ThN**

| Peak | Peak position (exp.) | ADE (exp.) | ADE (theo.) | Freq[a] (exp.) | Freq (theo.) | Final SO state |
|------|----------------------|------------|-------------|----------------|--------------|----------------|
| X | 1.576 (1) | 1.576 (1) | 1.55 | 944 | 960 | $^2\Sigma_{1/2}^+$ |
| a | 1.693 (1) | | | | | |
| b | 1.809 (1) | | | | | |

Both the computational ADE and vibrational frequencies were calculated at CCSD(T) level. AVQZ and VQZ-PP basis sets were used for N and Th, respectively.
[a]The value of the first vibrational interval $\Delta G_{1/2}$.

character of the canonical Kohn-Sham orbitals, we also carried out the NBO calculations to generate the Weinhold's natural localized molecular orbitals (NLMOs)[36,37]. As detailed shown in Supplementary Table 5, besides the apparent three bonding NLMOs for both ThC and ThN, there exists one additional NLMO which has albeit small, but non-negligible contributions from Th hybrid orbitals. These facts indicate that apart from the apparent one σ bond and two π bonds, there may exist a potential, albeit weak bond between Th and C/N.

## EDA-NOCV analysis

The combination of energy decomposition analysis (EDA) with the natural orbitals for chemical valence (NOCV) has been substantiated to be a powerful tool in interpreting the chemical bonding between two fragments through dividing the orbital interactions into pair-wise contributions. Within EDA-NOCV scheme, each decomposed inter-acting energy term is associated with a peculiar type of bond, which is ascribed by the visual survey of the shape of deformation density, and each component of the interaction term can also be quantified[38,39]. Owing to the fact that currently the EDA-NOCV is implemented within the framework of density functional theory, the reliability of EDA-NOCV calculations basically depends on the accuracy of the single-reference Kohn-Sham wavefunctions in describing the systems. In other words, the studied electronic state should not have strong multireference character. In order to confirm this assumption, we carried out the single point CASSCF (complete active space self-consistent field) calculations for the above-mentioned ground states

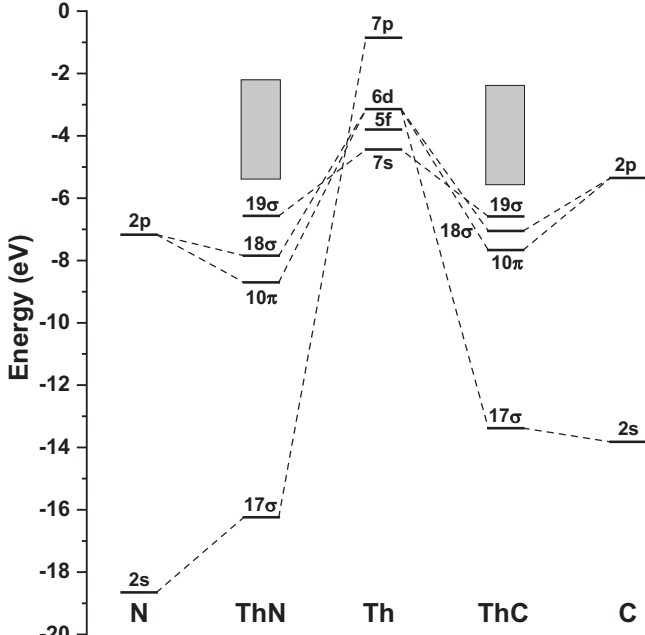

**Fig. 3 | MO energy-level diagrams of ThN and ThC.** The restricted open-shell Kohn-Sham MO energies from statistical SAOP calculations were directly used to depict the diagram. The electron configurations are ThC $(17\sigma)^2(10\pi)^4(18\sigma)^1(19\sigma)^1$ and ThN $(17\sigma)^2(10\pi)^4(18\sigma)^2(19\sigma)^1$, respectively. Source data are provided as a Source data file.

of ThC and ThN at the CCSD(T) optimized geometries, in which the same active spaces of SO-CASPT2 calculations were selected. Our calculations suggest that the total CAS wavefunction for the ground state of ThC can be expressed as the linear combination of configurations:

$$\Psi_{\text{ThC}} = 0.951(17\sigma)^2(10\pi)^4(18\sigma)^1(19\sigma)^1$$
$$- 0.196(17\sigma)^2(10\pi)^2(11\pi)^2(18\sigma)^1(19\sigma)^1 + \text{small terms}$$

and that of ThN can be expressed as:

$$\Psi_{\text{ThN}} = 0.956(17\sigma)^2(10\pi)^4(18\sigma)^2(19\sigma)^1$$
$$- 0.179(17\sigma)^2(10\pi)^2(11\pi)^2(18\sigma)^2(19\sigma)^1 + \text{small terms}$$

Thus, the leading configuration has the dominant contribution to the ground state CAS wavefunction, and the single-reference Kohn-Sham wavefunctions can somewhat reliably describe the ground state of the two studied neutral molecules.

We performed EDA-NOCV calculations for the ground state of ThC and ThN, in which ionic atoms are used as the interacting moieties (Table 3 and Fig. 4). We have to emphasize that in our selected scheme, one of the 7s electrons of the thorium has been taken away by the non-metallic elements. This setting is based on the fact that in both ThC and ThN, the singly occupied molecular orbital 19σ has a dominant Th-7s contribution. The theoretical atomic charges based on DFT calculations in Supplementary Table 6 also show that Th is positively charged in these two neutral molecules. Although the selected ionic reference states overestimate the energy terms related to electrostatic interaction and Pauli repulsion, as will be explicated in the following session, it can still show meaningful insights into the nature of orbital interaction between metallic and non-metallic atoms.

The EDA-NOCV results listed in Table 3 show that thorium has a larger interaction energy with nitrogen than carbon. The deformation densities representing the amount and shape of charge flow accompanying the bond formation are displayed in Fig. 4. Based on the shape

**Table 3 | EDA-NOCV results of ThC and ThN at the PBE/TZ2P level using ionic fragments as interacting moieties. All energy values are given in kcal/mol**

| Energy Term | Interacting Moieties | |
|---|---|---|
| | $Th^+ + C^-$ | $Th^+ + N^-$ |
| $\Delta E_{int}$ | −277.4 | −333.9 |
| $\Delta E_{Steric}$ | 20.4 | 69.5 |
| $\Delta E_{Pauli}$ | 494.5 | 751.0 |
| $\Delta E_{elstat}$ | −474.1 | −681.5 |
| $\Delta E_{orb}$ | −297.8 | −403.4 |
| $\Delta E_{\sigma1}$ | −90.3 (30.3%) | −182.3 (45.2%) |
| $\Delta E_{\pi}$ | −80.9 (27.2%) | −94.6 (23.5%) |
| $\Delta E_{\pi'}$ | −80.9 (27.2%) | −94.6 (23.5%) |
| $\Delta E_{\sigma2}$ | −34.1 (11.5%) | −23.1 (5.7%) |

of the deformation densities, the orbital interaction of both molecules can be decomposed into four pair-wise contributions, i.e., one dative Th←C/N σ bond, two electron-sharing Th-C/N π bonds, and one weak polarized Th←C/N σ bond. As shown in Fig. 4, the classification of interacting term as σ1 dative bond is mainly due to the fact that we select ionic fragments as interacting moieties ($Th^+ + N^-/C^-$). This interaction component, which is denoted as σ1 bond, has clearly C/N-to-Th density transfer for two studied neutral molecules. The net charge flow $\Delta\rho_{\alpha+\beta}$ of σ1 bond in ThC is 0.65 in the C→Th direction, and the interaction energy is −90.3 kcal/mol, which contributes 30.3% of $\Delta E_{orb}$. This result is consistent with the Kohn-Sham MOs that the 18σ bonding orbital is singly occupied in ThC. In ThN, the σ bonding energy is −182.3 kcal/mol, which provides 45.2% of $\Delta E_{orb}$. The deformation densities in Fig. 4 show that the sum of α and β contributions of σ1 bond has net charge flow with a value of 1.24 in the N→Th direction.

The combination of the two electron-sharing π bonds have an interacting energy of −182.3 kcal/mol, giving rise to the strongest contributions to the orbital interaction in ThC, and its total percentage in $\Delta E_{orb}$ is 54.4%. Therefore, in ThC the π bonds account for more than half of the covalent bonding energy based on our EDA-NOCV calculation. The π bonds in ThN also take the percentage of 47.0% in the total $\Delta E_{orb}$. The direction of the charge flow in π bonds, which is from thorium to the non-metallic elements, as illustrated in Fig. 4, indicates that in ThC and ThN, the π-backdonation plays a tantamount role in the stabilization of the chemical bond as the σ-donation.

The π bonds and σ1 bond comprise the classic picture of triple bond (with bond order 2.5 for ThC because the σ-bonding orbital is singly occupied). Apart from the three above-mentioned bonds corresponding to the classic picture of triple bond, one non-negligible component of the bonding, which features the polarization of the electron charge from C or N atom to Th, arises from the EDA-NOCV calculations. This weak σ-type bond has interaction energies of −34.1 and −23.1 kcal/mol for ThC and ThN, respectively. The sum $\Delta\rho_{\alpha+\beta}$ values of the deformation densities associated with this term for ThC and ThN are 0.54 and 0.40, respectively. Through the deformation densities and the charge flow of the bond denoted as σ2, we can recognize the accumulated charge density resembling the $d_z^2$ atomic orbital. This agrees well with the SFO population analyses that a certain amount of Th-$6d_z^2$ AO participated in 17σ MO. Thus, the charge flow from AOs of C/N towards the Th-$6d_z^2$ AO is involved in the stabilization of the diatomic thorium carbide and nitride, giving rise to the total bond order 3.5 and 4 for ThC and ThN, respectively.

To conclude, by combining the experimental and theoretical investigations of the electronic structures and chemical bonding in diatomic thorium carbide and nitride, we report the EA of these two neutral molecules as 1.562 and 1.576 eV, respectively. The anomalous short bond length and very large vibrational frequency of ThN have

been attributed to the unusual quadruple bond multiplicity of nitrogen. The quadruple bond Th≡N has been analyzed in detail and verified by a sequence of theoretical approaches, especially the EDA-NOCV method. Due to one of the σ-bond is singly occupied, the similar bonding situation of thorium carbide is assigned with BO as 3.5. This interesting result may open up new routes towards the design of molecules for studying multiple bonds and give insights to the understanding of the chemical interaction between Th and C/N in the Th-containing organometallic compounds.

## Methods
### Experiments
The experiment was done using the cryogenically slow-electron velocity-map imaging (SEVI) apparatus, which consists of a laser ablation ion source, a cold octupole radio-frequency (rf) ion trap, a time-of-flight mass spectrometer, a mass gate and a photoelectron velocity-map imaging system. The $ThN^-$ and $ThC^-$ anions were generated by the pulsed Nd:Y-Al-garnet laser ablation of a thorium metal disk in the presence of $NF_3$ and $CH_4$ gas, respectively. The hot anions lose their kinetic energy through collisions with the buffer gas (20% $H_2$ and 80% He) in the octupole radio-frequency (RF) ion trap, which temperature can be tuned in the range of 5K-300K by a liquid helium refrigerator. The stored anions are ejected out via pulsed potentials, and analyzed via time-of-flight mass spectrometry. In this work, the operating frequency of the laser was set at 20 Hz, the mixture of 20% $H_2$ and 80% He was used as the buffer gas, which was delivered by a pulsed valve, and the cold trap temperature was set at 15 K. The anions were stored and cooled in the ion trap for 45 ms and then ejected out via pulsed potentials on the end caps of the ion trap, and analyzed by a Wiley-McLaren type time of flight (TOF) mass spectrometer. The $ThN^-$ and $ThC^-$ anions were selected via a mass gate, which selected different qualities according to the delay time, before being photo-detached by probing laser from a Spectra-Physics dye laser system (400−920 nm, linewidth 0.06 $cm^{-1}$). The wavelength of the probing laser was monitored by a wavelength meter (HighFinesse WS6-600) with an accuracy of 0.02 $cm^{-1}$. Then the outgoing photoelectrons were guided to a detector consisting of a pair of microchannel plates and a fluorescent screen by the electric field of the velocity-map imaging system. The hitting positions, which are directly related to the photoelectrons, were recorded by a CCD camera. Since the distribution of outgoing photoelectrons had a cylindrical symmetry about the polarization axis, the photoelectron distribution could be reconstructed from the projected imaging via the maximum entropy velocity Legendre reconstruction method. The photoelectrons with the same velocity form a spherical shell, where the radius of the shell r is proportional to their velocity that is directly related to the kinetic energy $E_k = \alpha r^2$. The coefficient α could be determined by varying the probing photon energy $h\nu$. The corresponding binding energy (BE) of the detachment channel was extracted from BE = $h\nu − E_k$. A tunable laser with photon energy tuned slightly above the threshold was usually used to obtain high-resolution energy spectra.

### Theory and calculations
The geometries and the electronic structures of the studied molecules in this article were obtained from both the density functional theory (DFT) and high-level wave function theory (WFT) calculations. The DFT calculations were carried out using the exchange-correlation functional implemented in Amsterdam Density Functional (ADF 2016.01)[40] and Gaussian 16[41], and Molpro2020.2[42] was used to perform the sophisticated WFT calculations. The theoretical bond lengths of all the anions and neutral molecules were calculated at the CCSD(T) (coupled-cluster singles-and-doubles plus perturbative triples) level[43]. We also performed the SO-CASPT2[44] calculations to acquire the energies of the low-lying excited states of neutral ThC and ThN at each of their anion ground

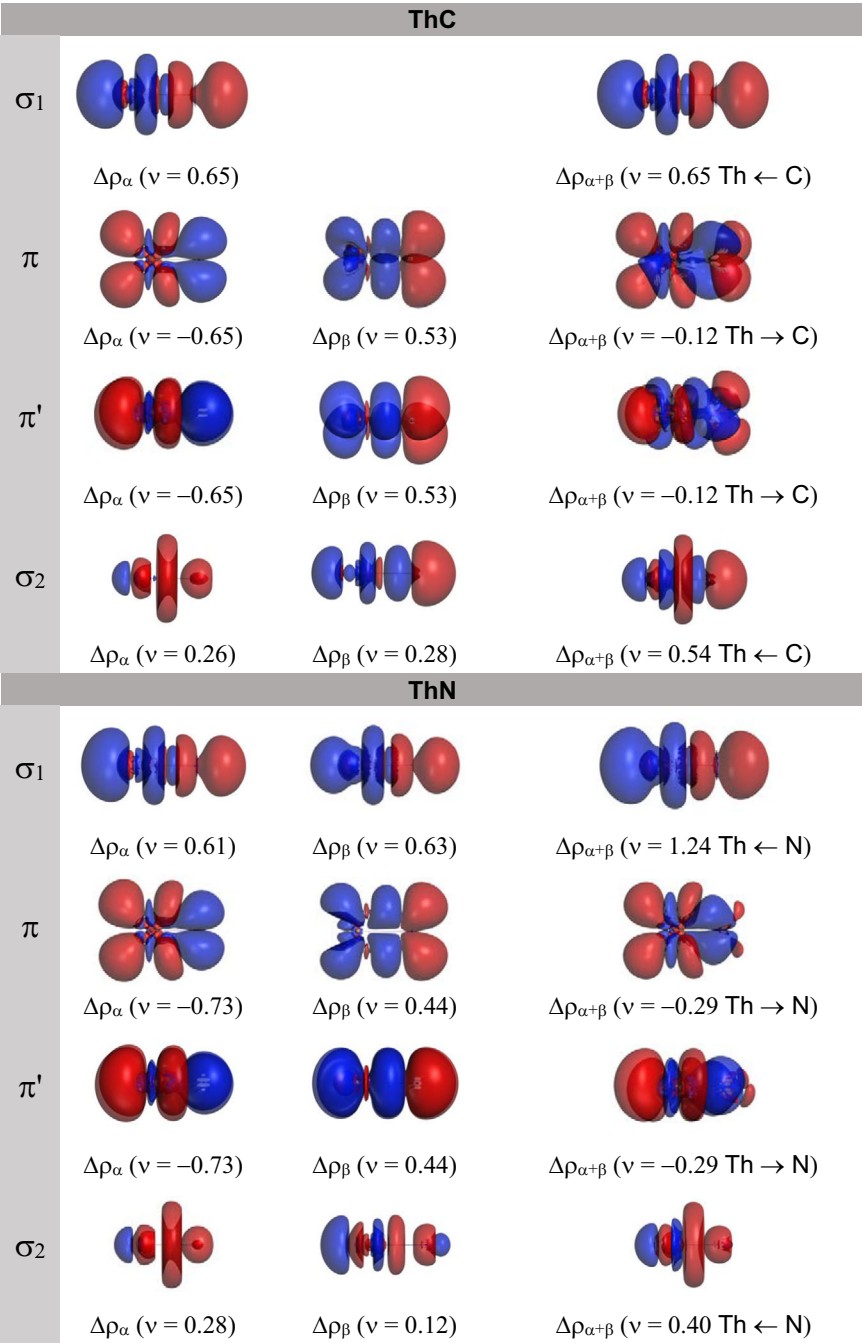

**Fig. 4 | Deformation densities and charge flow for bonds in ThC and ThN.** The charge flow is red → blue. The numerical value in the parentheses indicates the amount of deformation density Δρ.

state geometries, and the theoretical results were used to interpret the experimental spectrum. The Kohn-Sham wave functions generated at the DFT level were used to conduct the theoretical analyses to further investigate the nature of the chemical bonding. In Gaussian and Molpro calculations, the all-electron augmented valence triple-ζ basis sets aug-cc-pVTZ were employed for C and N[45], while the Stuttgart energy-consistent relativistic pseudopotentials ECP60MDF(Th)[46] and the corresponding cc-pVTZ basis sets[47] were used for Th. Slater basis sets with the quality of triple-ζ plus two polarization functions (TZ2P) were used in ADF calculations. In order to further reduce the basis set errors of sophisticated electron correlation method, we also carried out the CCSD(T) calculations using augmented quadruple-ζ basis sets aug-cc-pVQZ for C and N[45], quadruple-ζ basis sets cc-pVQZ-PP[46,47] for Th. The aug-cc-

pVXZ and cc-pVXZ-PP (X = T and Q) were abbreviated as AVXZ and VXZ-PP in this paper for convenience. Detailed information on the computational methods can be found in the Supplementary Information.

### Reporting summary
Further information on research design is available in the Nature Portfolio Reporting Summary linked to this article.

### Data availability
All data generated in this study are provided in the Source data file and Supplementary Information. Additional data supporting the findings of this study are available from the corresponding author upon request. Source data are provided with this paper.

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

## Acknowledgements

X.G.X. acknowledges support from National Natural Science Foundation of China (Grant No. 22273121), Science and Technology Program of Guangzhou, China (Grant No. 202102080374) and the Fundamental Research Funds for the Central Universities, Sun Yat-sen University (Grant No. 2021qntd12). Z.F. acknowledges support from National Natural Science Foundation of China (Grant No.21703281), Youth Innovation Promotion Association, Chinese Academy of Science (Grant No. 2021255). H.L. acknowledges support from National Natural Science Foundation of China (Grant No.21573273), the Strategic Priority Research Program of the Chinese Academy of Sciences (XDA02000100, XDA21000000), and the Program of Instrument and Equipment Development of Chinese Academy of Sciences (YJKYYQ20200050). C.N. acknowledges support from National Natural Science Foundation of China (Grant No.12374244, 11974199) and The National key R&D program of China (2018YFA0306504). C.D. acknowledges supports from National Natural Science Foundation of China (Grant No 22273065). The calculations were performed at in-house computing cluster and Tianhe-2 supercomputer located in the National Supercomputer Center in Guangzhou, China.

## Author contributions

Z.F., R.T., Y.L., C.H., Y.W., J.H. and C.D. performed the experimental studies. J.-Q.W., H.-S.H. and X.-G.X. carried out the computation and theoretical analysis. H.-S.H., X.-G.X., C.N., H.L. and J.L. supervised the work. X.-G.X. and H.L. conceptualized and wrote the draft of the manuscript. All authors subsequently reviewed and approved the final version of the manuscript.

## Competing interests

The authors declare no competing interests.
