## [Peer Review File · Nature Communications]

REVIEWER COMMENTS

Reviewer #1 (Remarks to the Author):

In this work, high-resolution anion photoelectron spectroscopy alongside high-level quantum chemical calculations are applied to two actinide-containing molecules, ThC and ThN. They interpret their findings as the first reported evidence that a nitrogen atom can participate in a quadruple bond. This is notable in the context of traditional views of bonding orders taught in any introductory-level chemistry course; the maximum bonding order for a main-group element is limited to three, due to the orbital structure of the atom. The methods used -- cryo-SEVI and high-level calculations -- are well-suited for an analysis of molecular properties, and the quality of the results seems to be consistent with the standards of the Nature family of journals.

However, as written, the paper is not up to the standards of Nature Communications. I would suggest the authors make an effort to improve the writing style and organization, keeping in mind that this is not a chemistry-specific journal, and that the average reader may not immediately recall many of the concepts that are taken as tacit. Some more concrete suggestions are as follows:

1. In the introduction, the authors need to do more to connect their quantitative results to the qualitative interpretation, as well as to clearly emphasize why this qualitative interpretation is novel. For example, it might be helpful to clarify *why* these rules for bond orders exist, rather than simply stating them as common knowledge. Further, when they mention that bond energies / lengths / vib. freqs. inform about the bond order, the authors should explicitly say why this makes sense (higher-order bonds are stronger, shorter, and stiffer).
2. The experimental method, cryo-SEVI, is not mentioned at all in the introduction, which makes the paper seem as though it is presenting purely theoretical results. The experiment and the relevant quantities it provides is information that should be made clear early on. Since it can give vibrational frequencies, it can be connected to the previous point, allowing the reader to understand why the approach used here is appropriate.
3. Figures 1 and 2 should be altered to be more informative for a reader that is not used to looking at electron images. At the very least, I would suggest including an energy diagram with the typical harmonic oscillator type energy levels that is commonly referred to when SEVI is presented to a more general audience (for example, Figure 1 in reference 16), with the transitions labelled to match the corresponding peaks. This would go a long way to connecting the different aspects of this work (theory/experiment and the interpretation of a quadruple bond). It also would help greatly for understanding the text in the Results section, which at present is a bit difficult to get through, particularly if you do not regularly read anion PES papers.
4. In figs 1-2, it also seems as though the caption does not quite correspond to the pictured figure, as it says that the laser polarization arrow is below the photoelectron image, whereas it is to the left. The authors should double-check that the captions are correct.
5. There are some places where the wording makes it impossible to determine what the authors are trying to say. For example, lines 101-104, "The lowest binding energy band X corresponds to the detachment transition from the ground state of ThN⁻ to that of neutral ThN, generates whereas the equally spacing binding energy bands X, A, and B, indicating the vibrational progression for the ground-state transition..." I have no idea what is trying to be communicated here.
6. While the discussion section contains a lot of insightful ideas, it could benefit from a clearer structure to help guide the reader through the author's thought process. Consider organizing the ideas into more distinct paragraphs and using headings or topic sentences to introduce each new point. This will help make the content more accessible to readers and improve the overall readability of the paper.

Reviewer #2 (Remarks to the Author):

I have read the work performed by Xiong's group with great interest. Recently, it has been shown that carbon and other main group elements can form quadruple bonds. This experimental-theoretical study provides clear evidence that nitrogen can also create such bonds, making it worthy of publication in Nature Communications. I have a few comments to address:

- 1) Please include in the discussion the recent publication by Frenking's group, which discusses a quadruple bond involving a fluorine (F) atom.
- 2) In the C2 discussion, please include a related reference: J. Chem. Phys. 2013, 138, 151102.
- 3) Regarding R(Th-C), I assume it should be r(Th-C). To avoid confusion, it is recommended to use Th-C as a subscript rather than in parentheses.
- 4) There have been documented issues with the triple zeta TZVP basis set. I suggest employing an aug-cc-pVTZ basis set, as the additional diffuse functions are essential for accurately describing the anion. Consider that this change could impact electronic distribution. If feasible, using a quadruple zeta basis set would be even more beneficial.
- 5) How is the steric component of the EDA (energy decomposition analysis) defined?
- 6) Does the dispersion term in DFT calculations affect the bond distance?
- 7) What is the value of effective bond order (EBO) value?

Overall, I believe this work is valuable and should be published with the abovementioned revisions.

Reviewer #3 (Remarks to the Author):

The Unusual Quadruple Bonding of Nitrogen in ThN

Xiao-Gen Xiong, Zejie Fei, Jia-Qi Wang, Rulin Tang, Yuzhu Lu, Changcai Han, Yongtian Wang, Jing Hong, Changwu Dong, Hanshi Hu, Chuan-Gang Ning, Hongtao Liu, and Jun Li

The authors report the anion photoelectron imaging spectra of ThN^- and ThC^- diatomic molecules, which provides the electron affinities of the neutral diatomics as well as vibrational frequencies of the neutrals. They compare these results with calculated energies and vibrational frequencies (DFT and ab initio calculations were done), then analyze the computed results to draw conclusions on the bond order of the two diatomics.

The photoelectron imaging spectra are of high quality, and the calculations are also fine. However, given the main focus on the electronic structure of neutral ThN, I do not see what new insights the authors provide when compared to the previous work on ThN reported by Heaven and coworkers. The new feature of the molecule the authors determine is the electron affinity and the photoelectron spectrum, to which they give little attention. That is, they do not perform simulations to determine whether the structures and ground states calculated for the anions map onto the experimental data. They oddly refer to the VDE calculations when comparing with their spectral origins, when the appropriate comparison between the calculated energies of the anion and neutral and peak "X" is the difference in the zero-point corrected energies of the fully optimized structures of the anions and neutrals. The authors do not compare their measured frequencies with the reported frequency of ThN diatomic (previously reported to be 950 cm^{-1}), which deviates significantly from their observed value. They also do not address what appears to be a vibrational or electronic hot band in their ThC^- spectrum.

A second concern is the characterization of polarization of charge density within a diatomic molecule as constituting a bond. At best, one might consider it a partial bond, so the claim of a quadruple bond strikes the reviewer as a stretch. The authors refer to the high vibrational frequency, short bondlength, and yes, they do indicate a strong Th-N bond. However, a curious reader might compare the N-N diatomic vibrational frequency, and scale the frequency by the root of the ratio of the reduced mass of N_2 and ThN, and conclude that the ThN bond is much weaker than the N_2 triple bond.

Other more easily addressable issues:

1. It is difficult to understand the meaning of some sentences because of the poor grammatical construction.
2. The authors, in the "methods" section, state that the radii of the annular images obtained using VMI are proportional to the electron kinetic energy, which is incorrect. The radii are proportional to the velocity.
3. Are the depictions in figure 4 directly relatable to the orbitals in the MO diagram in Figure 3? If so, why not refer to them in a way that makes it straightforward to compared the two figures?
4. The CAS wavefunctions for the ground state of ThC and ThN have 8 electrons. ThN should have one more electron than ThC.

REVIEWER COMMENTS

Reviewer #1 (Remarks to the Author):

In this work, high-resolution anion photoelectron spectroscopy alongside high-level quantum chemical calculations are applied to two actinide-containing molecules, ThC and ThN. They interpret their findings as the first reported evidence that a nitrogen atom can participate in a quadruple bond. This is notable in the context of traditional views of bonding orders taught in any introductory-level chemistry course; the maximum bonding order for a main-group element is limited to three, due to the orbital structure of the atom. The methods used -- cryo-SEVI and high-level calculations -- are well-suited for an analysis of molecular properties, and the quality of the results seems to be consistent with the standards of the Nature family of journals.

However, as written, the paper is not up to the standards of Nature Communications. I would suggest the authors make an effort to improve the writing style and organization, keeping in mind that this is not a chemistry-specific journal, and that the average reader may not immediately recall many of the concepts that are taken as tacit. Some more concrete suggestions are as follows:

1. In the introduction, the authors need to do more to connect their quantitative results to the qualitative interpretation, as well as to clearly emphasize why this qualitative interpretation is novel. For example, it might be helpful to clarify **why** these rules for bond orders exist, rather than simply stating them as common knowledge. Further, when they mention that bond energies / lengths / vib. freqs. inform about the bond order, the authors should explicitly say why this makes sense (higher-order bonds are stronger, shorter, and stiffer).

Thanks for this valuable suggestion, we have revised the manuscript and added several sentences to explain why we say that maximum bond order for main group elements is generally three, and also, we have mentioned the relation between order with bond length, strength, etc.

2. The experimental method, cryo-SEVI, is not mentioned at all in the introduction, which makes the paper seem as though it is presenting purely theoretical results. The experiment and the relevant quantities it provides is information that should be made clear early on. Since it can give vibrational frequencies, it can be connected to the previous point, allowing the reader to understand why the approach used here is appropriate.

Thanks for the suggestions. In order to emphasize the importance of the experiments, we explicitly added the contents (Page 2 of the revised MS, also displayed below) to clarify that advanced experimental techniques play a vital role in the study of fundamental aspects of chemistry:

“Using advanced gas phase spectroscopy, we can experimentally determine the bond strength by probing the vibrational frequencies between the bounded atoms. For examples, with the help of anion photoelectron spectroscopy, the Rh≡B quadruple bond has been found and rationalized in RhB(BO)⁻ and RhB¹³. Later on another boron-metal quadruple bond was found in BFe(CO)₃⁻¹⁴ by infrared photodissociation spectroscopy. Inspired by these previous reports, herein we explore the

potential and bonding characteristics of nonmetallic elements, C and N, in the second period, to form a quadruple bond with heavy element Th in ThC and ThN molecules, respectively. Using the sophisticated quantum chemical method, the calculated bond length in ThC is comparable with $R_{(\text{Th}=\text{C})}$ predicted by the average triple bond radii, while the bond length in ThN is noticeably shorter than predicted $R_{(\text{Th}=\text{N})}$. In order to further study these two molecules, the cryogenically slow velocity map imaging (cryo-SEVI) experiments were carried out to characterize the nature of the considered systems, and the comprehensive theoretical calculations present unprecedented high bond orders of 3.5 and 4 for ThC and ThN, respectively.”

3. Figures 1 and 2 should be altered to be more informative for a reader that is not used to looking at electron images. At the very least, I would suggest including an energy diagram with the typical harmonic oscillator type energy levels that is commonly referred to when SEVI is presented to a more general audience (for example, Figure 1 in reference 16), with the transitions labelled to match the corresponding peaks. This would go a long way to connecting the different aspects of this work (theory/experiment and the interpretation of a quadruple bond). It also would help greatly for understanding the text in the Results section, which at present is a bit difficult to get through, particularly if you do not regularly read anion PES papers.

Thanks for the kindly suggestion, we have revised the two figures as follows:

Fig. 1 (a) Photoelectron images and (b) spectra for ThC^- at 1.909 eV, (c) the energy level Schematic for the ground state $\text{ThC}^- (^2\Sigma_0^+) \rightarrow \text{ThC} (^3\Sigma_0^+ + ^3\Sigma_1^+)$ transition and first excited state $\text{ThC}^- (^2\Sigma_0^+) \rightarrow \text{ThC} (^1\Sigma_0^+)$ transition processes, the transitions are labelled to match the corresponding peaks in (b). The vertical lines in (b) represent vibrational structures, and the double arrow on the left of the image in (a) indicates the laser polarization.

Fig. 2 (a) Photoelectron images and (b) spectra for ThN⁻ at 1.909 eV, (c) the energy level Schematic for the ground state ThN⁻ (¹Σ₀⁺) → ThN (²Σ_{1/2}⁺) transition process, the transitions are labelled to match the corresponding peaks in (b). Vertical lines in (b) represent vibrational structures, and the double arrow on the left of the image in (a) indicates the laser polarization.

4. In figs 1-2, it also seems as though the caption does not quite correspond to the pictured figure, as it says that the laser polarization arrow is below the photoelectron image, whereas it is to the left. The authors should double-check that the captions are correct.

Yes, the laser polarization arrow should be revised as “the double arrow on the left of the image in (a) indicates the laser polarization” in Fig.1 and Fig.2.

5. There are some places where the wording makes it impossible to determine what the authors are trying to say. For example, lines 101-104, "The lowest binding energy band X corresponds to the detachment transition from the ground state of ThN⁻ to that of neutral ThN, generates whereas the equally spacing binding energy bands X, A, and B, indicating the vibrational progression for the ground-state transition..." I have no idea what is trying to be communicated here.

Thanks for these comments, we rewrite the paragraph and revised the label accordingly, the caption X and A represent the electronic state configuration, while a, b, c and d denote the excited vibrational states. All these modified contents have been marked in yellow in the file with track changes.

6. While the discussion section contains a lot of insightful ideas, it could benefit from a clearer structure to help guide the reader through the author's thought process. Consider organizing the ideas into more distinct paragraphs and using headings or topic sentences to introduce each new point. This will help make the content more accessible to readers and improve the overall readability of the paper.

We have followed this kindly suggestion and added the headings (using bold fonts and highlighted in yellow) to make the paragraph more readable.

Reviewer #2 (Remarks to the Author):

I have read the work performed by Xiong's group with great interest. Recently, it has been shown that carbon and other main group elements can form quadruple bonds. This experimental-theoretical study provides clear evidence that nitrogen can also create such bonds, making it worthy of publication in Nature Communications. I have a few comments to address:

1) Please include in the discussion the recent publication by Frenking's group, which discusses a quadruple bond involving a fluorine (F) atom.

Thanks for pointing out this missing reference, we have included it (Ref. 7) in the revised manuscript.

2) In the C2 discussion, please include a related reference: J. Chem. Phys. 2013, 138, 151102.

Thanks! We have included this reference (Ref. 5) in the introduction part. "Recently, with the development of the high-level theoretical calculations, the possibility of quadruple bonding between two carbon atoms has been discussed in C2 molecule²⁻⁵".

3) Regarding R(Th-C), I assume it should be r(Th-C). To avoid confusion, it is recommended to use Th-C as a subscript rather than in parentheses.

Following the reviewer's kindly suggestion, we have replaced R(Th-C), R(Th-N), R(Th=C), R(Th≡N) as R_(Th-C), R_(Th-N), R_(Th=C), R_(Th≡N), respectively.

4) There have been documented issues with the triple zeta TZVP basis set. I suggest employing an aug-cc-pVTZ basis set, as the additional diffuse functions are essential for accurately describing the anion. Consider that this change could impact electronic distribution. If feasible, using a quadruple zeta basis set would be even more beneficial.

We totally agree with the reviewer on this point. Generally, accurate description of anion needs diffuse functions, and the electron correlation methods, especially the coupled-cluster theory, are very sensitive to the size of the basis sets. Therefore, we tried our best to do the affordable calculations to make the theoretical results as accurate as possible.

Other than basis sets used in the previous submitted manuscript, we performed the additional calculations at the CCSD(T) level. As we couldn't find the optimized augmented/diffuse basis functions at the level of aug-cc-pVTZ-PP for Th, and Th is generally positively charged in the metal carbide/nitride, therefore we only use cc-pVTZ-PP and cc-pVQZ-PP for Th. In total, there are three types of basis sets we have used for CCSD(T) calculations:

- (a) cc-pVTZ-PP for Th, cc-pVTZ for C and N;
- (b) cc-pVTZ-PP for Th, aug-cc-pVTZ for C and N;
- (c) cc-pVQZ-PP for Th, aug-cc-pVQZ for C and N.

We listed the optimized geometries, vibrational frequencies and 1st ADEs (difference in the zero-point corrected energies of the fully optimized structures of the anions and neutrals, as Reviewer #3

suggested) for the studied anions and neutral molecules as follows:

The optimized geometries:

	$R_{(\text{Th-C})}/\text{\AA}$		$R_{(\text{Th-N})}/\text{\AA}$	
	ThC ⁻	ThC	ThN ⁻	ThN
CCSD(T) ^a	1.986	1.951	1.857	1.822
CCSD(T) ^b	1.984	1.947	1.858	1.823
CCSD(T) ^c	1.985	1.948	1.854	1.820

- cc-pVTZ-PP for Th, cc-pVTZ for C and N;
- cc-pVTZ-PP for Th, aug-cc-pVTZ for C and N;
- cc-pVQZ-PP for Th, aug-cc-pVQZ for C and N.

The vibrational frequencies:

	Freq. (cm ⁻¹)			
	ThC ⁻	ThC	ThN ⁻	ThN
CCSD(T) ^a	771	843	900	956
CCSD(T) ^b	771	846	900	959
CCSD(T) ^c	773	847	904	960

- cc-pVTZ-PP for Th, cc-pVTZ for C and N;
- cc-pVTZ-PP for Th, aug-cc-pVTZ for C and N;
- cc-pVQZ-PP for Th, aug-cc-pVQZ for C and N.

The theoretical values of ADE₁ (in eV):

	ThC ⁻	ThN ⁻
CCSD(T) ^a	1.49	1.51
CCSD(T) ^b	1.51	1.54
CCSD(T) ^c	1.52	1.55

- cc-pVTZ-PP for Th, cc-pVTZ for C and N;
- cc-pVTZ-PP for Th, aug-cc-pVTZ for C and N;
- cc-pVQZ-PP for Th, aug-cc-pVQZ for C and N.

From the above three tables, we can see that the geometries, vibrational frequencies and energies all have the systematical convergency from (a) to (c), as it should be. So, the best strategy for the revised manuscript should be that we replace all the theoretical values with the new calculated results using quadruple- ζ basis sets to acquire best accuracy. But unfortunately, when we performed the CASPT2 calculations with quadruple- ζ basis sets (aug-cc-pVQZ for C and N, cc-pVQZ-PP for Th), we met the technical issues that we haven't solved yet. The main problem is that our CASPT2 calculations use a very large space, the extra addition of the basis-functions makes the calculations intractable.

Although the CASPT2 is also sensitive with the size of the basis sets, based on the results from the CCSD(T) calculations we carried out, we can make a reasonable guess that CASPT2 calculations

employing triple- ζ level basis sets should give results with acceptable accuracy for the titled molecules.

At this stage, we unlikely to make a compromise. In the revised manuscript, for the geometries, vibrational frequencies and ADEs, we include the CCSD(T) calculations using both triple- ζ (VTZ-PP for Th, AVTZ for C and N) and quadruple- ζ (VQZ-PP for Th, AVQZ for C and N) basis sets; for the energies of excited states, we only performed the SO-CASPT2 employing triple- ζ basis sets, but the more accurate geometries generated at the level of CCSD(T) employing quadruple- ζ (VQZ-PP for Th, AVQZ for C and N) basis sets were used. From the Table 1 and Table 2 of the revised manuscript, compared with the experiment values, we can see this combination gives reasonably accurate results.

As the DFT is not as sensitive as post-HF methods with the size of basis sets, and from Table S1 we can see the calculations using triple- ζ basis sets can generate acceptable bond lengths, therefore we only report the results of DFT calculations employing triple- ζ basis sets (diffuse functions have been added for C and N).

5) How is the steric component of the EDA (energy decomposition analysis) defined?

For the energy decomposition, we used the definition from the paper "Kohn-Sham Density Functional Theory: Predicting and Understanding Chemistry" (*Rev. Comput. Chem.*; Lipkowitz, K. B. and Boyd, D. B., Eds.; Wiley-VCH: New York, 2000, Vol. 15, 1-86.), in which the interaction energy between two atoms or two fragments can be decomposed into the sum of electrostatic interactions, Pauli repulsion, and the orbital interaction energy:

$$\Delta E_{\text{bonding}} = \Delta E_{\text{electrostatic}} + \Delta E_{\text{Pauli}} + \Delta E_{\text{orbint}} = \Delta E_{\text{steric}} + \Delta E_{\text{orbint}}$$

The steric component is defined as the sum of the electrostatic interaction energy and Pauli repulsion.

6) Does the dispersion term in DFT calculations affect the bond distance?

Thanks for this reminder, we did forget to consider the possible effects of dispersion towards the bond length in the DFT calculations. Therefore, we checked this issue and performed the DFT calculations using corrections of Grimme's D3BJ dispersion (Grimme S., Ehrlich S. and Goerigk L., *J. Comput. Chem.*, 2011, 32, 1456–1465) to optimize the geometries of the studied molecules. Luckily, as it is shown in the following table, and also the higher bond order generally suggesting strong interactions between the two bonding atoms, we argue that we can safely neglect the effects of the dispersion term towards the bond lengths.

	$R_{(\text{Th}-\text{C})}/\text{\AA}$		$R_{(\text{Th}-\text{N})}/\text{\AA}$	
	ThC ⁻	ThC	ThN ⁻	ThN
PBE	1.971	1.934	1.850	1.819
PBE-D3BJ	1.971	1.934	1.850	1.819
PBE0	1.959	1.923	1.832	1.802
PBE0-D3BJ	1.959	1.923	1.832	1.802

B3LYP	1.977	1.939	1.848	1.816
B3LYP-D3BJ	1.977	1.939	1.848	1.816
TPSS	1.976	1.938	1.850	1.820
TPSS-D3BJ	1.976	1.938	1.850	1.820

7) What is the value of effective bond order (EBO) value?

The EBO values of ThC and ThN generated from our CASSCF calculations are 2.41 and 2.91, respectively.

Overall, I believe this work is valuable and should be published with the abovementioned revisions.

Thank you so much! We really appreciate all your valuable comments and suggestions.

Reviewer #3 (Remarks to the Author):

comment part 1: The authors report the anion photoelectron imaging spectra of ThN⁻ and ThC⁻ diatomic molecules, which provides the electron affinities of the neutral diatomics as well as vibrational frequencies of the neutrals. They compare these results with calculated energies and vibrational frequencies (DFT and ab initio calculations were done), then analyze the computed results to draw conclusions on the bond order of the two diatomics.

The photoelectron imaging spectra are of high quality, and the calculations are also fine. However, given the main focus on the electronic structure of neutral ThN, I do not see what new insights the authors provide when compared to the previous work on ThN reported by Heaven and coworkers. The new feature of the molecule the authors determine is the electron affinity and the photoelectron spectrum, to which they give little attention.

Reply: Thanks for the very useful comments. It is well known that Heaven's group have been dedicated in actinide molecule research for decades and they reported many very high-quality spectroscopies of actinide diatomic molecule (including ThN/ThN⁺). Therefore, in this work, we were initially intended focus on the bonding analysis, and reported the newest EAs from anion photoelectron spectroscopy. However, inspired by reviewer's comments and also benefitted from the high resolution of cryo-SEVI, we take a more sophisticated data evaluation and some other valuable molecule constants can also be obtained. Thereafter we revised the manuscript accordingly.

The details are as follows: (1) Figure 1 and Figure 2 have been reorganized, since the measured vibrational intervals exhibit anharmonicities for both ThN and ThC, and we try to fit the data with the first order anharmonic constant to obtain the more reliable diatomic constants, the new reported frequencies for ThC and ThN are 815 and 944 cm⁻¹, respectively (as displayed in Table 1 and Table 2). (2) Anion vibrational interval can be estimate from the measured vibrational hotband (peak hb in figure 1) for ThC⁻, and the value is 710 cm⁻¹ as shown in Table 1.

comment part 2: That is, they do not perform simulations to determine whether the structures and ground states calculated for the anions map onto the experimental data. They oddly refer to the VDE calculations when comparing with their spectral origins, when the appropriate comparison between the calculated energies of the anion and neutral and peak "X" is the difference in the zero-point corrected energies of the fully optimized structures of the anions and neutrals.

Simulations are performed for the spectra assignment, and the VDE was replaced with EA, and we have calculated the values based on "difference in the zero-point corrected energies of the fully optimized structures of the anions and neutrals", as the reviewer kindly suggested.

comment part 3: The authors do not compare their measured frequencies with the reported frequency of ThN diatomic (previously reported to be 950 cm⁻¹), which deviates significantly from their observed value. They also do not address what appears to be a vibrational or electronic hot band in their ThC⁻ spectrum.

We rewrite the experimental results, and add the hb data analysis in the revised manuscript. As stated

in the reply for comment part 1, the revised experimental frequency for ThN (944 cm^{-1}) is very close to the previously reported value.

comment part 4: A second concern is the characterization of polarization of charge density within a diatomic molecule as constituting a bond. At best, one might consider it a partial bond, so the claim of a quadruple bond strikes the reviewer as a stretch. The authors refer to the high vibrational frequency, short bondlength, and yes, they do indicate a strong Th-N bond. However, a curious reader might compare the N-N diatomic vibrational frequency, and scale the frequency by the root of the ratio of the reduced mass of N₂ and ThN, and conclude that the ThN bond is much weaker than the N₂ triple bond.

We agree that some people may argue that it is equivocal to say the “polarization of charge density” is actually a chemical bond. In some sense, the concept of coordinate covalent bond (donation and back-donation) which is common in the organometallic chemistry, is also a kind of “intensive polarization”, or more precisely, charge transfer. We are also open that it may deserve more discussions in this issue.

For the scaling of the frequency by the reduced mass, we argue that it may not be wise to judge the orders solely by comparing these values. We have to notice that ThN has a large bond length than N₂, therefore the features of the bonding curve around the minimum should be different. Take the N₂ and P₂ for example, if we compare the values of the force constants, N₂ is much larger than P₂, but we cannot deny that P₂ has triple bond.

1. It is difficult to understand the meaning of some sentences because of the poor grammatical construction.

We take the advice from the #1 reviewer, and reorganized the discussion with headings for each part. And we also polished the revised manuscript carefully and rewrite many sentences.

2. The authors, in the “methods” section, state that the radii of the annular images obtained using VMI are proportional to the electron kinetic energy, which is incorrect. The radii are proportional to the velocity.

The description in this part is corrected as “The photoelectrons with the same velocity form a spherical shell, where the radius of the shell r is proportional to their velocity that directly related to the kinetic energy.”

3. Are the depictions in figure 4 directly relatable to the orbitals in the MO diagram in Figure 3? If so, why not refer to them in a way that makes it straightforward to compared the two figures?

The depictions (deformation densities) in Figure 4 are not directly related to the orbitals in Figure 3. The deformation densities representing the amount and shape of charge flow accompanying the bond formation, while the MOs in Figure 3 is the Kohn-Sham molecular orbitals. They are two different sets of objects.

4. The CAS wavefunctions for the ground state of ThC and ThN have 8 electrons. ThN should have one more electron than ThC.

Thanks so much for pointing out this mistake (typo). We have corrected the typo in the revised manuscript.

REVIEWERS' COMMENTS

Reviewer #1 (Remarks to the Author):

I appreciate the authors' efforts to address my previous suggestion regarding broadening the introduction to engage a wider audience of Nature Communications. However, there are a few concerns I'd like to address.

Firstly, there are sentences in the revised introduction that appear somewhat confusing. For instance, the sentence "we can experimentally determine the bond strength by probing the vibrational frequencies between the bounded atoms" might be clearer if rephrased as "we can experimentally assess bond strengths by measuring vibrational frequencies of diatomic molecules."

Secondly, there is a consistent use of "bounded" instead of "bonded," which deviates from common terminology.

Furthermore, the sentence from lines 39-41 regarding the tetrahedral arrangement of valence shell electrons is unclear and does not effectively support the argument for a maximum bond order of three. The argument presented in the abstract regarding an octet is comprehensive; adding a reference to a general chemistry textbook would further enhance the point.

I also noticed the phrase "Using the sophisticated quantum chemical method" on line 67 lacks specificity. It might be helpful to provide more detail about the method being referred to, or alternatively phrase it as "Using a sophisticated quantum chemical method."

On a positive note, the expanded energy level diagrams in Figures 1 and 2 significantly aid in interpreting the spectra. Likewise, the reorganization of the discussion section is a notable improvement.

In conclusion, I believe the results presented are valuable for publication as they contribute to the collection of high-resolution spectral information. However, I must mention that there are several typographical and language issues that need addressing before the paper can be considered for publication. These issues appear predominantly in the introduction. It's my recommendation that the authors meticulously review and refine the introduction to ensure its alignment with the rest of the technically sound paper.

Reviewer #2 (Remarks to the Author):

I have just a couple of points to address. Firstly, in the references section, please ensure that all authors are listed individually rather than using "et al." Consequently, references 1, 5, 16, and 32 should be updated accordingly.

Secondly, could you kindly include the values of the T1 diagnostic? If these values happen to be relatively high, it would be beneficial to provide justification for employing DFT or even CCSD(T) in your analysis.

Reviewer #3 (Remarks to the Author):

Upon review of the revised manuscript, in addition to reading the response document and the revised supporting information, it is clear the authors have carefully considered the reviewer comments, addressed concerns on the scientific content of the manuscript.

There remain major issues with the actual manuscript composition. I will leave it to the editors to determine whether this manuscript can be edited to be accessible to experts in the area of small molecule spectroscopy, and, hopefully, those outside of this immediate area of expertise.

As an example, in the introduction, the authors appropriately responded to a different reviewer's suggestion that it was not immediately obvious in the introduction that experimental results figured prominently in the study with the following:

In order to further study these two molecules, the cryogenically slow velocity map imaging (cryoSEVI) experiments were carried out to characterize the nature of the considered systems, and the comprehensive theoretical calculations present unprecedented high bond orders of 3.5 and 4 for ThC and ThN, respectively.

It is laudable that the authors addressed the reviewer's concern, but the sentence will be difficult to digest for anyone who is not familiar with cryogenically-cooled slow... and how the community reconciles cryo-SEVI spectra with theoretical results.

I can add a large number of examples like this, including the way in which the authors describe a vibrational progression, and how they define how they compute the ADE.

Overall, I again assert that the spectra are of high quality, and in response to other reviewer comments, the theoretical component has been strengthened. The authors did not raise the nuance of how a bond is defined particularly deeply in this revised manuscript, and it is still not suitable for publication in Nature Chem in its current form.

REVIEWER COMMENTS

Reviewer #1 (Remarks to the Author):

I appreciate the authors' efforts to address my previous suggestion regarding broadening the introduction to engage a wider audience of Nature Communications. However, there are a few concerns I'd like to address.

Firstly, there are sentences in the revised introduction that appear somewhat confusing. For instance, the sentence "we can experimentally determine the bond strength by probing the vibrational frequencies between the bounded atoms" might be clearer if rephrased as "we can experimentally assess bond strengths by measuring vibrational frequencies of diatomic molecules."

Thank you so much and we have followed the kindly suggestion to avoid the confusion.

Secondly, there is a consistent use of "bounded" instead of "bonded," which deviates from common terminology.

We adopted the common terminology and replaced the "bounded" with "bonded" in the revised manuscript.

Furthermore, the sentence from lines 39-41 regarding the tetrahedral arrangement of valence shell electrons is unclear and does not effectively support the argument for a maximum bond order of three. The argument presented in the abstract regarding an octet is comprehensive; adding a reference to a general chemistry textbook would further enhance the point.

The octet rule says the electronic structures of compounds require the valence shell of main group element has 8 electrons. We think octet rule cannot guarantee the maximum bond order of the main group element is 3. Take the carbon as an example, carbon atom has four valence electrons and can form 4 bonds with other atoms, but generally these 4 bonds are formed with different atoms (and there are some exceptions, e.g., C_2 can be one of them). Therefore, we insist on keeping the argument of "tetrahedral arrangement" in the article, and in order to avoid the misapprehension and obscurity, we rephrased it as the editor suggest:

"This is based on the postulation that the tetrahedral arrangement of the four pairs of electrons in the valence shells of the main-group element, so that a maximum of three pairs of electrons can form a bond when two tetrahedra are sharing a face."

I also noticed the phrase "Using the sophisticated quantum chemical method" on line 67 lacks specificity. It might be helpful to provide more detail about the method being referred to, or alternatively phrase it as "Using a sophisticated quantum chemical method."

Thanks for pointing out this issue, in the revised manuscript we explicitly stated the level of theory we used to acquire the most accurate bond length in our study.

On a positive note, the expanded energy level diagrams in Figures 1 and 2 significantly aid in interpreting the spectra. Likewise, the reorganization of the discussion section is a notable improvement.

In conclusion, I believe the results presented are valuable for publication as they contribute to the collection of high-resolution spectral information. However, I must mention that there are several typographical and language issues that need addressing before the paper can be considered for publication. These issues appear predominantly in the introduction. It's my recommendation that the authors meticulously review and refine the introduction to ensure its alignment with the rest of the technically sound paper.

Thank you so much, we have done our best to revise the manuscript and eliminate the blemish in the introduction.

Reviewer #2 (Remarks to the Author):

I have just a couple of points to address. Firstly, in the references section, please ensure that all authors are listed individually rather than using "et al." Consequently, references 1, 5, 16, and 32 should be updated accordingly.

Thanks for pointing out the inconsistency in the reference section. We have listed all the authors in the revised manuscript.

Secondly, could you kindly include the values of the T1 diagnostic? If these values happen to be relatively high, it would be beneficial to provide justification for employing DFT or even CCSD(T) in your analysis.

We have checked the T1 diagnostic values of our CCSD(T) calculations using quadruple- ζ basis sets (aug-cc-pVTZ for C & N, cc-pVTZ-PP for Th), and the results are listed in the following table:

	ThC ⁻	ThC	ThN ⁻	ThN
T1	0.02979755	0.0296255	0.02641781	0.01909322

The rule of thumb suggests the CCSD(T) results can be trusted if the T1 diagnostic values less than 0.02 for closed shell systems and 0.03 for radicals. As all the four molecules in our studies are open-shell, although the T1 diagnostic values are very closed to the upper limit of the "safe" value, we still argue that the CCSD(T) results (at least the energies and geometries) are reasonable for the ground states of the studied molecules. The T1 diagnostic values combined with the compositions of the CAS wave functions stated in the manuscript all suggest that both ThC and ThN has some multireference character, but not that much. Hence, we think the single-reference methods, including the DFT and CCSD(T), can be cautiously used to predict the reliable ground state properties of these two systems.

Reviewer #3 (Remarks to the Author):

Upon review of the revised manuscript, in addition to reading the response document and the revised supporting information, it is clear the authors have carefully considered the reviewer comments, addressed concerns on the scientific content of the manuscript.

There remain major issues with the actual manuscript composition. I will leave it to the editors to determine whether this manuscript can be edited to be accessible to experts in the area of small molecule spectroscopy, and, hopefully, those outside of this immediate area of expertise.

As an example, in the introduction, the authors appropriately responded to a different reviewer's suggestion that it was not immediately obvious in the introduction that experimental results figured prominently in the study with the following:

In order to further study these two molecules, the cryogenically slow velocity map imaging (cryo-SEVI) experiments were carried out to characterize the nature of the considered systems, and the comprehensive theoretical calculations present unprecedented high bond orders of 3.5 and 4 for ThC and ThN, respectively.

It is laudable that the authors addressed the reviewer's concern, but the sentence will be difficult to digest for anyone who is not familiar with cryogenically-cooled slow... and how the community reconciles cryo-SEVI spectra with theoretical results.

I can add a large number of examples like this, including the way in which the authors describe a vibrational progression, and how they define how they compute the ADE.

Overall, I again assert that the spectra are of high quality, and in response to other reviewer comments, the theoretical component has been strengthened. The authors did not raise the nuance of how a bond is defined particularly deeply in this revised manuscript, and it is still not suitable for publication in Nature Chem in its current form.

Thanks for pointing out the possible difficulties for the potential readers. We have added the necessary sentence to explain the cryo-SEVI, and for other points we also try to make it more accessible for people outside the small molecule community. We added sentences in the Introduction section to explain the definition of single-, double- and triple-bond, which generally count the number of electron pairs that the two bonding atoms shared.

We added the definition of cryo-SEVI at its first point of use, refined the sentences which describe the vibrational progression and the way we compute the ADE has also been revised. We have done our best, to hopefully address all the concerns pointed out by the reviewers and the editor.

REVIEWER COMMENTS

Reviewer #1 (Remarks to the Author):

I appreciate the authors' efforts to address my previous suggestion regarding broadening the introduction to engage a wider audience of Nature Communications. However, there are a few concerns I'd like to address.

Firstly, there are sentences in the revised introduction that appear somewhat confusing. For instance, the sentence "we can experimentally determine the bond strength by probing the vibrational frequencies between the bounded atoms" might be clearer if rephrased as "we can experimentally assess bond strengths by measuring vibrational frequencies of diatomic molecules."

Thank you so much and we have followed the kindly suggestion to avoid the confusion.

Secondly, there is a consistent use of "bounded" instead of "bonded," which deviates from common terminology.

We adopted the common terminology and replaced the "bounded" with "bonded" in the revised manuscript.

Furthermore, the sentence from lines 39-41 regarding the tetrahedral arrangement of valence shell electrons is unclear and does not effectively support the argument for a maximum bond order of three. The argument presented in the abstract regarding an octet is comprehensive; adding a reference to a general chemistry textbook would further enhance the point.

The octet rule says the electronic structures of compounds require the valence shell of main group element has 8 electrons. We think octet rule cannot guarantee the maximum bond order of the main group element is 3. Take the carbon as an example, carbon atom has four valence electrons and can form 4 bonds with other atoms, but generally these 4 bonds are formed with different atoms (and there are some exceptions, e.g., C_2 can be one of them). Therefore, we insist on keeping the argument of "tetrahedral arrangement" in the article, and in order to avoid the misapprehension and obscurity, we rephrased it as the editor suggest:

"This is based on the postulation that the tetrahedral arrangement of the four pairs of electrons in the valence shells of the main-group element, so that a maximum of three pairs of electrons can form a bond when two tetrahedra are sharing a face."

I also noticed the phrase "Using the sophisticated quantum chemical method" on line 67 lacks specificity. It might be helpful to provide more detail about the method being referred to, or alternatively phrase it as "Using a sophisticated quantum chemical method."

Thanks for pointing out this issue, in the revised manuscript we explicitly stated the level of theory we used to acquire the most accurate bond length in our study.

On a positive note, the expanded energy level diagrams in Figures 1 and 2 significantly aid in interpreting the spectra. Likewise, the reorganization of the discussion section is a notable improvement.

In conclusion, I believe the results presented are valuable for publication as they contribute to the collection of high-resolution spectral information. However, I must mention that there are several typographical and language issues that need addressing before the paper can be considered for publication. These issues appear predominantly in the introduction. It's my recommendation that the authors meticulously review and refine the introduction to ensure its alignment with the rest of the technically sound paper.

Thank you so much, we have done our best to revise the manuscript and eliminate the blemish in the introduction.

Reviewer #2 (Remarks to the Author):

I have just a couple of points to address. Firstly, in the references section, please ensure that all authors are listed individually rather than using "et al." Consequently, references 1, 5, 16, and 32 should be updated accordingly.

Thanks for pointing out the inconsistency in the reference section. We have listed all the authors in the revised manuscript.

Secondly, could you kindly include the values of the T1 diagnostic? If these values happen to be relatively high, it would be beneficial to provide justification for employing DFT or even CCSD(T) in your analysis.

We have checked the T1 diagnostic values of our CCSD(T) calculations using quadruple- ζ basis sets (aug-cc-pVTZ for C & N, cc-pVTZ-PP for Th), and the results are listed in the following table:

	ThC ⁻	ThC	ThN ⁻	ThN
T1	0.02979755	0.0296255	0.02641781	0.01909322

The rule of thumb suggests the CCSD(T) results can be trusted if the T1 diagnostic values less than 0.02 for closed shell systems and 0.03 for radicals. As all the four molecules in our studies are open-shell, although the T1 diagnostic values are very closed to the upper limit of the "safe" value, we still argue that the CCSD(T) results (at least the energies and geometries) are reasonable for the ground states of the studied molecules. The T1 diagnostic values combined with the compositions of the CAS wave functions stated in the manuscript all suggest that both ThC and ThN has some multireference character, but not that much. Hence, we think the single-reference methods, including the DFT and CCSD(T), can be cautiously used to predict the reliable ground state properties of these two systems.

Reviewer #3 (Remarks to the Author):

Upon review of the revised manuscript, in addition to reading the response document and the revised supporting information, it is clear the authors have carefully considered the reviewer comments, addressed concerns on the scientific content of the manuscript.

There remain major issues with the actual manuscript composition. I will leave it to the editors to determine whether this manuscript can be edited to be accessible to experts in the area of small molecule spectroscopy, and, hopefully, those outside of this immediate area of expertise.

As an example, in the introduction, the authors appropriately responded to a different reviewer's suggestion that it was not immediately obvious in the introduction that experimental results figured prominently in the study with the following:

In order to further study these two molecules, the cryogenically slow velocity map imaging (cryo-SEVI) experiments were carried out to characterize the nature of the considered systems, and the comprehensive theoretical calculations present unprecedented high bond orders of 3.5 and 4 for ThC and ThN, respectively.

It is laudable that the authors addressed the reviewer's concern, but the sentence will be difficult to digest for anyone who is not familiar with cryogenically-cooled slow... and how the community reconciles cryo-SEVI spectra with theoretical results.

I can add a large number of examples like this, including the way in which the authors describe a vibrational progression, and how they define how they compute the ADE.

Overall, I again assert that the spectra are of high quality, and in response to other reviewer comments, the theoretical component has been strengthened. The authors did not raise the nuance of how a bond is defined particularly deeply in this revised manuscript, and it is still not suitable for publication in Nature Chem in its current form.

Thanks for pointing out the possible difficulties for the potential readers. We have added the necessary sentence to explain the cryo-SEVI, and for other points we also try to make it more accessible for people outside the small molecule community. We added sentences in the Introduction section to explain the definition of single-, double- and triple-bond, which generally count the number of electron pairs that the two bonding atoms shared.

We added the definition of cryo-SEVI at its first point of use, refined the sentences which describe the vibrational progression and the way we compute the ADE has also been revised. We have done our best, to hopefully address all the concerns pointed out by the reviewers and the editor.